# Recognition of tumor cells by Dectin-1 orchestrates innate immune cells for anti-tumor responses

Shiho Chiba[1,2†], Hiroaki Ikushima[1,2,3†], Hiroshi Ueki[1,2†], Hideyuki Yanai[1,2,3,4], Yoshitaka Kimura[1,2], Sho Hangai[1,2], Junko Nishio[1,2,4], Hideo Negishi[1,2,4], Tomohiko Tamura[5], Shinobu Saijo[6], Yoichiro Iwakura[7], Tadatsugu Taniguchi[1,2,3,4*]

[1]Department of Molecular Immunology, Institute of Industrial Science, The University of Tokyo, Tokyo, Japan; [2]Center for International Research on Integrative Biomedical Systems, Institute of Industrial Science, The University of Tokyo, Tokyo, Japan; [3]Max Planck-The University of Tokyo Center for Integrative Inflammology, Tokyo, Japan; [4]Japan Science and Technology Agency, Core Research for Evolution Science and Technology, Tokyo, Japan; [5]Department of Immunology, Yokohama City University Graduate School of Medicine, Yokohama, Japan; [6]Department of Molecular Immunology, Medical Mycology Research Center, Chiba University, Chiba, Japan; [7]Center for Animal Disease Models, Research Institute for Biomedical Sciences, Tokyo University of Science, Noda, Japan

*For correspondence: tada@m.u-tokyo.ac.jp

†These authors contributed equally to this work and are listed in alphabetical order

**Abstract** The eradication of tumor cells requires communication to and signaling by cells of the immune system. Natural killer (NK) cells are essential tumor-killing effector cells of the innate immune system; however, little is known about whether or how other immune cells recognize tumor cells to assist NK cells. Here, we show that the innate immune receptor Dectin-1 expressed on dendritic cells and macrophages is critical to NK-mediated killing of tumor cells that express N-glycan structures at high levels. Receptor recognition of these tumor cells causes the activation of the IRF5 transcription factor and downstream gene induction for the full-blown tumoricidal activity of NK cells. Consistent with this, we show exacerbated in vivo tumor growth in mice genetically deficient in either Dectin-1 or IRF5. The critical contribution of Dectin-1 in the recognition of and signaling by tumor cells may offer new insight into the anti-tumor immune system with therapeutic implications.

## Introduction

In recent years, our understanding of innate immune recognition of microbial components and its critical role in host defense against infection has grown considerably. The discovery of Toll-like receptors (TLRs) and other classes of signal-transducing innate immune receptors exemplifies a model of pathogen recognition by germline-encoded pattern-recognition receptors (PRRs) of the innate immune system that detect components of invading pathogens, termed pathogen-associated molecular patterns (PAMPs) (*Janeway and Medzhitov, 2002*; *Blasius and Beutler, 2010*; *Kawai and Akira, 2010*, *2011*). Upon recognition of PAMPs, these receptors activate NF-κB, interferon regulatory factors (IRFs) and other transcription factors, which then induce the transcription of cognate target genes (*Tamura et al., 2008*; *Ikushima et al., 2013*). This results in the evocation of innate immune responses that also instruct subsequent innate immune responses. Dead or injured cells also recruit and activate innate inflammatory cells in the absence of infection through recognition by PRRs of released self-derived molecules termed damage-associated molecular patterns (DAMPs) (*Bianchi, 2007*; *Rubartelli and Lotze, 2007*). To date, the role of innate immune receptors in anti-tumor innate immune response is unknown.

**eLife digest** When cells in the body grow and divide uncontrollably, cancerous tumors can form. An individual's likelihood of recovering from cancer is highly variable and often depends on the type of cancer and the extent of the disease at the start of treatment. Researchers are therefore interested in discovering how the body responds against cancerous cells.

The first line of defense against infection and disease is the body's innate immune system, which includes a suite of immune cells known as white blood cells. These cells patrol the body's organs and tissues in an effort to immediately respond to pathogens and damaged, stressed or otherwise abnormal host cells. Among white blood cells, natural killer cells are involved in identifying and destroying tumor cells. However, it was unclear whether or how other immune cells might help natural killer cells to destroy tumors. In addition, although immune cells detect pathogens and injured cells by producing proteins called pattern recognition receptors, it was unknown whether these receptors also detect tumor cells.

Here, Chiba et al. reveal that two other types of immune cell—dendritic cells and macrophages—play essential roles in helping natural killer cells to prevent tumors from growing in mice. The dendritic cells and macrophages produce a pattern recognition receptor called Dectin-1 that recognizes a molecule found on the surface of some—but not all—types of tumor cell. In doing so, Dectin-1 activates a critical signaling pathway and directs the activity of the natural killer cells so that they can effectively kill tumor cells. Chiba et al. found that these tumors grew faster in mice that lack the Dectin-1 protein.

The findings of Chiba et al. may also help to explain the effectiveness of certain antibodies—proteins that recognize and neutralize foreign objects such as bacteria and viruses—in cancer therapy. In addition, the Dectin-1 pathway presents a new avenue of research that may offer new cancer treatments.

It has been well-established that natural killer (NK) cells are essential effector cells of the innate arm of the immune system to control virus infections and tumor developments by exerting cytotoxicity of target cells (*Yokoyama and Plougastel, 2003*). NK cells are sub-classified into circulating conventional NKs (cNKs) and tissue-resident NKs (trNKs) (*Sojka et al., 2014*). Thus far, most of the functional studies on NK cells have focused on cNKs, which are found to be abundant in the spleen. In view of the critical role of cNKs (referred to as NKs hereafter unless stated otherwise) in anti-tumor innate immunity, the interesting issue for whether PRRs expressed on NK and/or other immune cells may recognize tumor cells for the enhancement of tumoricidal activity of NK cells. It has been reported that cell-to-cell contact between dendritic cells (DCs) and resting NK cells causes an enhancement of NK cell-mediated cytolytic activity against tumor cells (*Fernandez et al., 1999*); however, the involvement of PRRs in this context remains unknown. Of note, some TLRs appear to have the opposite effect (*Rakoff-Nahoum and Medzhitov, 2009*; *Pradere et al., 2014*). For instance, activation of TLR2 and TLR6 in macrophages enhances metastasis of tumor cells, wherein receptor activation is mediated by tumor cell-derived versican, an extracellular matrix proteoglycan that is up-regulated in many tumor cells (*Kim et al., 2009*).

During our studies on the IRF family of transcription factors in the context of regulation of PRR signaling and oncogenesis, we observed that mice genetically deficient for IRF5 (IRF5-deficient mice) show extensive lung metastasis of the B16F1 melanoma cells (hereafter referred to as B16 cells), a consequence reported to be controlled by NK cells. Since IRF5 is known to be activated by TLRs and other classes of PRRs, we asked whether any of these PRRs is involved in this process in the context of NK cell-mediated control of tumor cells. In light of the tumor-promoting effect by TLRs (*Rakoff-Nahoum and Medzhitov, 2009*; *Pradere et al., 2014*), we hypothesized that another class of PRRs expressed on the cell surface may be involved in the recognition of tumor cells for the NK cell-mediated anti-tumor response. Dectin-1, a C-type lectin receptor (CLR) family member known to recognize β-glucans of fungal cell wall (*Brown, 2006*), was of particular interest since (i) it is highly expressed in macrophages and DCs (*Herre et al., 2004*) and (ii) IRF5 is involved in Dectin-1 signaling for anti-fungal innate immune response (*del Fresno et al., 2013*).

We show here that Dectin-1 expressed on DCs and macrophages critically contributes to the enhancement of NK-mediated killing of tumor cells. Further, we demonstrate that IRF5 is activated by

Dectin-1 signaling in these immune cells and that this Dectin-1-IRF5 pathway constitutes a critical limb for their orchestration of NK cells. We also provide evidence that tumor cell-mediated Dectin-1 signaling is instigated by receptor recognition of N-glycan structures on the surface of some but not all tumor cells, which we propose to term tumor-associated molecular patterns (TAMPs). Finally, the in vivo significance of these observations is validated by the massive growth of the TAMP-expressing tumors in mice genetically deficient for either Dectin-1 or IRF5.

We discuss our results revealing a new facet of the CLR family of receptors in the orchestration of anti-tumor innate immune responses as well as future prospects of innate recognition and control of tumor cells, which may have clinical implications.

## Results

### A requirement for IRF5 in innate immune responses against B16 melanoma

Within the IRF family of transcription factors, several members including IRF3, IRF5 and IRF7 are particularly well-known for the pivotal roles they serve in the innate immune receptor-mediated gene induction programme (*Honda and Taniguchi, 2006*). Here, we challenged mice deficient in these individual transcription factors with the lung metastasis model of B16F1 melanoma cells (hereafter referred to as B16 unless stated otherwise). As shown in *Figure 1A,B*, we observed markedly enhanced metastasis of B16 cells in the lungs of mutant mice deficient in the *Irf5* gene (hereafter IRF5-deficient mice). Massive tumor growth was also observed when B16 cells were subcutaneously injected into the mutant mice (*Figure 1—figure supplement 1*).

To examine the contribution of immune cells, we next conducted bone marrow transplantation and found that IRF5 expression specifically in bone marrow-derived cells critically contributes to the suppression of the tumor cell metastasis (*Figure 1—figure supplement 2A*). Since the marked B16 lung metastasis was not observed in mice deficient in T and B lymphocyte development (*Figure 1—figure supplement 2B*), we next examined the contribution of IRF5 in anti-tumor innate immune response, wherein NK cells are best known for their direct anti-tumor cytotoxicity (*Smyth et al., 2002*). When whole splenocytes, of which about 3% are resting NK cells in both wild-type (WT) and IRF5-deficient mice (*Takaoka et al., 2005*), were subjected to an in vitro killing assay (*Brunner et al., 1968*) ($^{51}$Cr release assay) for B16 cells, a notable decrease of killing activity was observed in those from IRF5-deficient mice (*Figure 1C*; left panel). Interestingly, however, when NK cells were purified from these splenocytes and then subjected to the killing assay, the lack of IRF5 did not affect the NK activity (*Figure 1C*; middle panel). Since the killing activity is entirely abrogated by NK cell depletion in both WT and mutant splenocytes (*Figure 1C*; right panel), these results indicate there is a critical contribution of non-NK cells within the total splenocyte population for which IRF5 is critical for the full-blown killing activity of NK cells.

Consistent with this notion, when purified NK cells were co-cultured with CD11c$^+$ DCs and macrophage-enriched CD11b$^+$ cells from splenocytes (hereafter referred to as co-culture assay), a dose-dependent and significant increase in NK killing activity was observed, whereas T or B lymphocytes had no such effect (*Figure 1D*). Of note, a marked reduction of the NK killing activity was observed when CD11b$^+$ or CD11c$^+$ cells from IRF5-deficient mice were used in lieu of those from WT mice in the co-culture assay (*Figure 1—figure supplement 3*). The enhancement of NK killing activity was also made in another co-culture assay, wherein CD8$^+$CD11c$^+$, CD8$^-$CD11c$^+$, or CD11c$^-$CD11b$^+$ cells purified from splenocytes were used, indicating that both DCs and macrophages participate in the enhancement of NK cell-mediated killing activity (*Figure 1—figure supplement 4*).

### Contribution of the Dectin-1-IRF5 pathway in NK cell-mediated anti-tumor response

The above observations prompted us to investigate the contribution of innate immune receptor(s) that recognizes B16 cells and activate IRF5 for the NK-mediated killing of the tumor cells. Although IRF5 is best known as a transcription factor activated by TLR signaling, we observed that splenocytes from mice deficient in MyD88, the common adapter for all TLRs, shows killing activity against B16 cells at a rate even slightly higher than those from WT mice (*Figure 2—figure supplement 1*). We then focused our attention on Dectin-1, a member of the CLR family, which is expressed on the cell surface of innate immune cells. In fact, Dectin-1, known to recognize β-glucans of fungal cell wall (*Brown, 2006*), is widely expressed by DCs and macrophages and the activation of IRF5 by the Dectin-1-Syk pathway

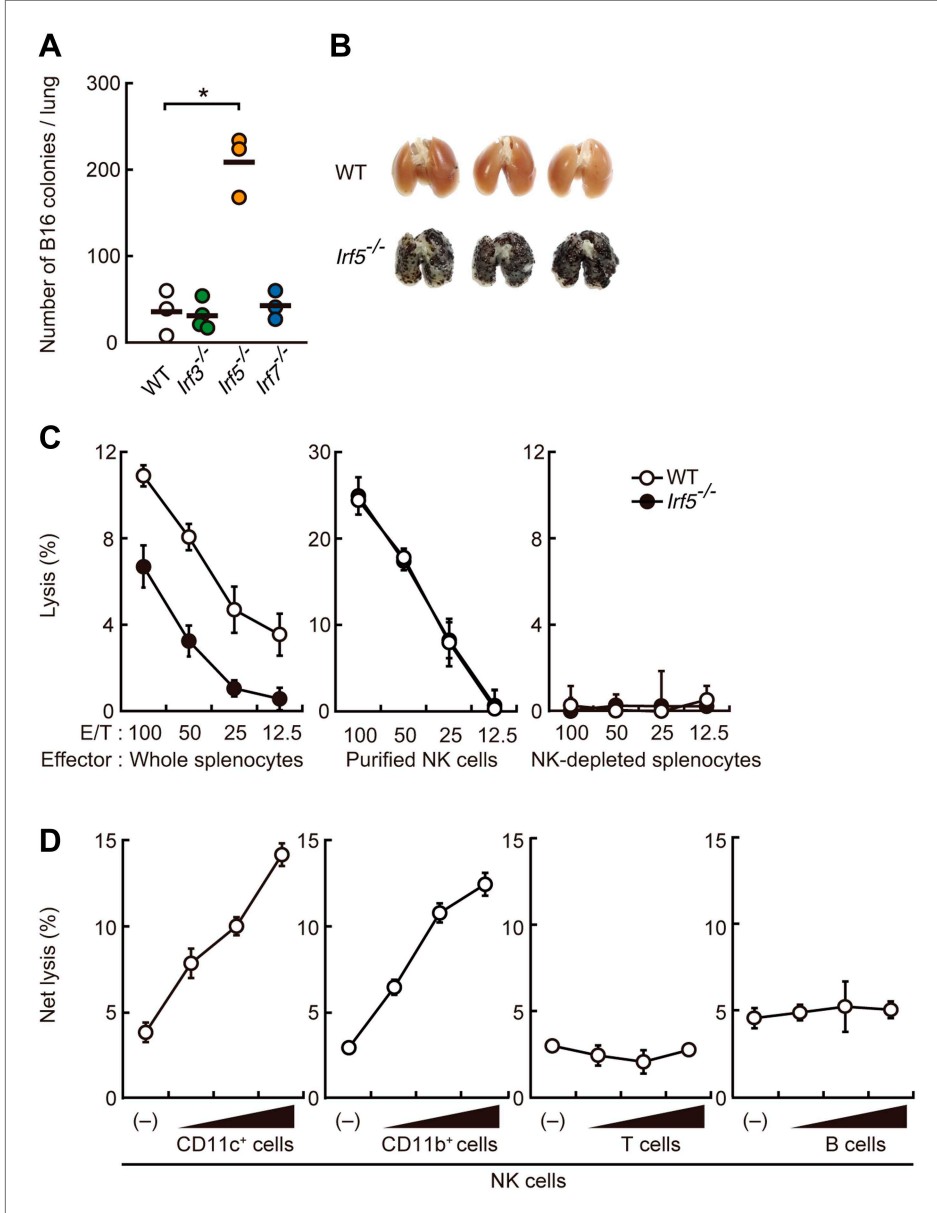

**Figure 1**. Critical contribution of IRF5 to the enhancement of NK cell-mediated anti-tumor responses. (**A**) Selective contribution of IRF5 in the suppression of lung metastasis of B16F1 cells. Number of metastasized colonies in lungs from wild-type (WT), *Irf3*[−/−], *Irf5*[−/−], or *Irf7*[−/−] mice 14 days after intravenous injection of $1 \times 10^6$ of B16F1 cells. Means are indicated as black bars. *p < 0.05 by Student's *t* test. (**B**) Representative images of lungs from WT or *Irf5*[−/−] mice 14 days after intravenous injection of $2 \times 10^6$ of B16F1 cells. (**C**) In vitro killing assay of immune cells from WT or *Irf5*[−/−] mice against B16F1 cells. Whole splenocytes (left panel), purified NK cells (middle panel), or NK-depleted splenocytes (right panel) from WT or *Irf5*[−/−] mice are mixed with $^{51}$Cr-labeled target B16F1 cells at the indicated ratios. 4 hr later, $^{51}$Cr radioactivity released from target cells was monitored. E/T: effector/target cell ratio. (**D**) Purified NK cells (WT; $1 \times 10^5$ cells) without or with $1 \times 10^5$, $2 \times 10^5$, or $4 \times 10^5$ of WT splenic CD11c[+], CD11b[+], T, or B cells were subjected to in vitro killing assay for B16F1 cells. Target cell lysis was measured by co-culturing target cells and myeloid cells, with (total values) or without NK cells (background values), and background values were subtracted from the total values. Each background lysis was less than 6% of maximum release. The calculated percentage of cytotoxicity was represented as Net lysis (%). In all in vitro killing assays, $1 \times 10^4$ of $^{51}$Cr-labeled B16F1 cells were used (**C** and **D**). All in vitro killing assays were performed at least three times with high reproducibility. Represented as means ± SD.

*Figure 1. Continued on next page*

*Figure 1. Continued*

The following figure supplements are available for figure 1:

**Figure supplement 1**. Critical role of IRF5 in the suppression of tumor growth.

**Figure supplement 2**. Requirement of IRF5 in myeloid cells for the suppression of tumor metastasis.

**Figure supplement 3**. Involvement of IRF5 in CD11b[+] and CD11c[+] cells to the enhancement of NK cell-mediated anti-tumor responses.

**Figure supplement 4**. Contribution of DCs and macrophages to the NK cell-mediated tumor killing.

has been implicated in the induction of type I interferon-β (IFN-β) gene upon fungal infection (*del Fresno et al., 2013*; *Leibundgut-Landmann et al., 2008*; *Backer et al., 2008*; *Brown et al., 2002*; *Honda and Taniguchi, 2006*).

We first examined whether IRF5 is activated in splenocytes through recognition of and signaling induced by B16 cells in a Dectin-1-dependent manner. Interestingly, we observed nuclear translocation of IRF5, a hallmark of its activation (*Honda and Taniguchi, 2006*), following incubation of WT splenocytes with B16 cells (in 50 to 1 or 100 to 1 cell ratio), but not in splenocytes from mice deficient in the *Clec7a* gene that encodes Dectin-1 (hereafter Dectin-1-deficient mice), indicating there is a Dectin-1 signal-dependent IRF5 activation by the tumor cells (*Figure 2A*). Of note, IRF5 activation by B16 cells was not inhibited by FK506, suggesting there is a calcineurin-independent pathway for this activation (*Figure 2—figure supplement 2*). As such, these data support a Dectin-1 signal-dependent IRF5 activation by tumor cells.

## Requirement of Dectin-1 in DCs and macrophages for in vitro tumoricidal activity of NK cells

Consistent with the above notion, a dramatic reduction of in vitro tumoricidal activity was observed with Dectin-1-deficient splenocytes (*Figure 2B*). When NK cells were purified from WT and Dectin-1-deficient splenocytes and subjected to the in vitro killing assay for B16 cells, a slight decrease in cell killing activity of Dectin-1-defcient NK cells was seen as compared to WT NK cells, indicating an ancillary role of the B16-Dectin-1 signaling in NK cells (*Figure 2—figure supplement 3A*). *Dectin-1* mRNA expression was indeed observed in NK cells and other innate immune cells (*Figure 2—figure supplement 3B,C*); however, Dectin-1 expression in NK cells does not account for the above Dectin-1-IRF5 axis as IRF5-deficient NK cells show normal cell killing activity (*Figure 1C*; middle panel).

When WT-derived resting NK cells were purified and subjected to the co-culture assay, a marked reduction of NK killing activity was also observed when Dectin-1-deficient CD11b[+] or CD11c[+] cells were used in lieu of those from WT mice (*Figure 2C*). It is worth noting that, consistent with the results showing the existence of IRF5-independent pathway for the Dectin-1 signaling, the reduction of NK cell killing activity was even more pronounced when CD11b[+] or CD11c[+] cells from Dectin-1-deficient mice were subjected to the co-culture assay compared to those from IRF5-deficient mice (*Figure 1—figure supplement 3*). These observations in toto underscore the involvement of Dectin-1 on DCs and/or macrophages in the full-blown NK-mediated tumoricidal activity. On the other hand, since the NK cell-enhancing activity is not totally abrogated by the absence of Dectin-1 in CD11b[+] or CD11c[+] cells (*Figure 2C*), there is perhaps contribution of additional innate immune receptor(s) in these cells for NK cell activation ('Discussion').

## Loss of tumor growth control in Dectin-1-deficient mice

On the basis of the above observations, we examined the in vivo contribution of Dectin-1 to anti-tumor innate immune responses by challenging WT and Dectin-1-deficient mice with the B16 cell lung metastasis model. As shown in *Figure 2D,E*, a marked enhancement of metastasis of B16 cells was seen in the lungs of Dectin-1-deficient mice as compared with those of WT mice. Loss of tumor growth control also occurred when Dectin-1-deficient mice were inoculated subcutaneously with B16 cells (*Figure 2—figure supplement 4*).

Of note, the metastasis to the lung was significantly more profound than that observed in IRF5-deficient mice (*Figure 2—figure supplement 5*); this observation is consistent with the above in vitro

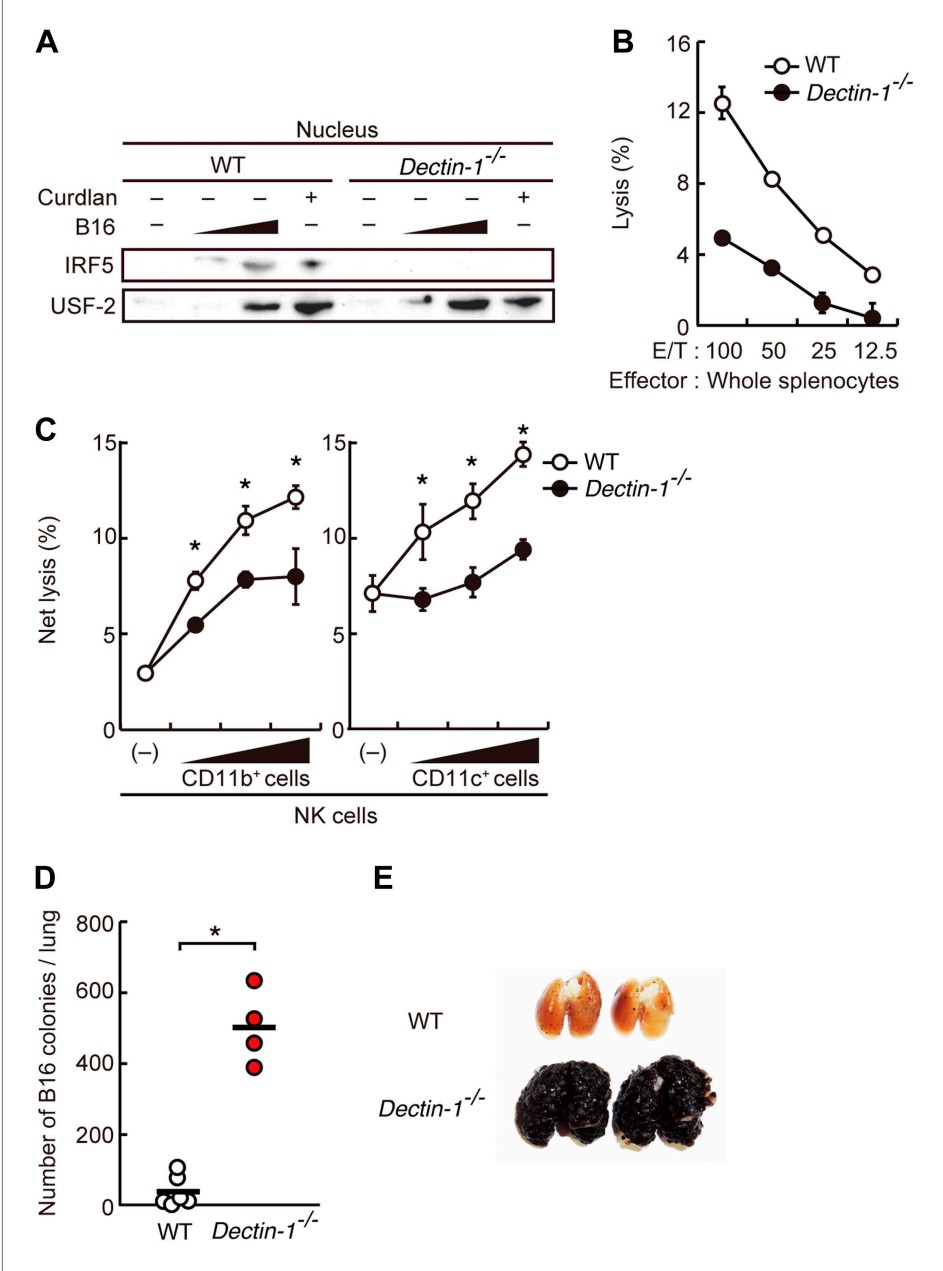

**Figure 2**. Critical role of Dectin-1 signaling in DCs and macrophages for IRF5 activation by and NK cell-mediated killing against B16F1 cells. (**A**) Nuclear translocation of IRF5 in WT or *Dectin-1*[−/−] splenocytes ($5 \times 10^7$ cells) co-incubated with B16F1 cells ($5 \times 10^5$ or $1 \times 10^6$ cells) or stimulated with curdlan (30 µg/ml) for 6 hr. Nuclear protein fraction from the culture was analyzed by immunoblotting for IRF5 and USF-2. USF-2 was used as a nuclear marker protein. (**B**) In vitro killing activity of whole splenocytes from WT or *Dectin-1*[−/−] mice against B16F1 cells. (**C**) In vitro killing activity of purified NK cells (WT; $1 \times 10^5$ cells) against B16F1 cells was assessed in the absence or presence of $1 \times 10^5$, $2 \times 10^5$, or $4 \times 10^5$ of WT or *Dectin-1*[−/−] splenic CD11b[+] (left panel) or CD11c[+] (right panel) cells. The percentage of cytotoxicity was calculated as noted in the legend of *Figure 1D* and represented as Net lysis (%). Each background lysis was less than 6% of maximum release. In all in vitro killing assays, $1 \times 10^4$ of [51]Cr-labeled B16F1 cells were used (**B** and **C**). All in vitro killing assays were performed at least three times with high reproducibility. Represented as means ± SD. *p < 0.05 by Student's *t* test. Of note, we could not found mRNA induction for typical inflammatory and cytotoxic mediators in the co-culture system, wherein the ratio of B16F1 cells, DCs, and NK cells is 1:30:10 (*Figure 2—figure supplement 8*). As such, from these analyses, it seems unlikely that Dectin-1 signaling in DCs affects the expression of these molecules in NK cells. (**D**) Number of metastasized colonies in
*Figure 2. Continued on next page*

*Figure 2. Continued*

lungs from WT or *Dectin-1$^{-/-}$* mice intravenously injected with $1 \times 10^6$ of B16F1 cells. Means are indicated as black bars. *p < 0.05 by Student's *t* test. (**E**) Representative images of lungs from WT or *Dectin-1$^{-/-}$* mice 14 days after intravenous injection of $2 \times 10^6$ of B16F1 cells.

The following figure supplements are available for figure 2:

**Figure supplement 1**. Dispensable role of MyD88 in anti-tumor killing activity of NK cells.

**Figure supplement 2**. Effect of FK506 treatment on IRF5 activation in splenocytes.

**Figure supplement 3**. Minor effects of Dectin-1 expressed in NK cells on the tumor killing activity.

**Figure supplement 4**. Essential role of Dectin-1 in the suppression of tumor growth.

**Figure supplement 5**. Contribution of Dectin-1 and IRF5 to the control of tumor metastasis.

**Figure supplement 6**. Dispensable role of Dectin-1 in NK-cell-independent tumor suppression.

**Figure supplement 7**. Normal population and functions of Dectin-1-deficient NK cells.

**Figure supplement 8**. Expression levels of cytotoxic mediators and inflammatory cytokines in co-culture system.

data, which shows that the IRF5 pathway contributes partly to Dectin-1 signaling (*Figure 2C*, *Figure 1—figure supplement 3*). In addition, the control of lung metastasis of B16 cells is equally lost between WT and Dectin-1-deficient mice when NK cells were depleted prior to tumor challenge (*Figure 2—figure supplement 6*); this result further supports the notion that NK cells are indeed the effector cells and that they need the assistance of DCs and macrophages for which Dectin-1 signaling is critical, in the orchestration for the innate control of tumor cells in vivo.

It is worth noting that no overt difference was observed between WT and Dectin-1-deficient mice in terms of (i) the proportion of splenic NK cells, (ii) production of IFN-γ by NK cells in vitro, and (iii) NK cell-dependent clearance of murine cytomegalovirus (MCMV) in vivo (*Figure 2—figure supplement 7*). Therefore, NK cell-dependent immune responses are not globally impaired by the absence of Dectin-1; impairment is selective to the anti-tumor response.

## Requirement of N-glycan structures for the recognition of Dectin-1

Does Dectin-1 recognize a molecular structure(s) on B16 cells? To address this question, we generated a soluble form of Dectin-1 conjugated to human IgG1 Fc (termed sDectin-1) to detect binding of Dectin-1 to the cell surface. Interestingly, substantial binding of sDectin-1 to B16 cells was detected, whereas binding was almost undetectable to mouse embryonic fibroblasts (MEFs) and other primary, non-transformed cells (*Figure 3A*, *Figure 3—figure supplement 1*).

Since β-glucans, known ligands of Dectin-1, are not expressed by mammalian cells (*Brown, 2006*), we hypothesized the presence of other types of glycosylated structure(s) on B16 cells that are mediating the recognition by Dectin-1. In this context, it has been shown that enhanced glycosylation levels provide growth advantages to many if not all tumor cells (*Granovsky et al., 2000*; *Gu and Taniguchi, 2008*). Interestingly, we found that sDectin-1 binding to B16 cells was markedly reduced upon N-glycosidase treatment, while treatment by O-glycosidase or neuraminidase showed only a marginal effect, suggesting there is a major requirement for N-glycan structures to Dectin-1 binding (*Figure 3B*, *Figure 3—figure supplement 2*). Consistent with this, a marked reduction of tumoricidal activity of splenocytes was observed when B16 cells were pretreated with N-glycosidase, whereas O-glycosidase treatment did not affect this activity (*Figure 3C*). Further, N-glycosidase treatment of B16 cells did not affect the in vitro killing activity of purified NK cells (*Figure 3—figure supplement 3*), indicating in toto there is a critical role of Dectin-1 recognition of and signaling by N-glycan structures on tumor cells by DCs and macrophages.

We then investigated the nature of N-glycan structures on B16 cells by subjecting the supernatant generated after N-glycosidase treatment to mass spectrometric (MS) analysis. As reported previously,

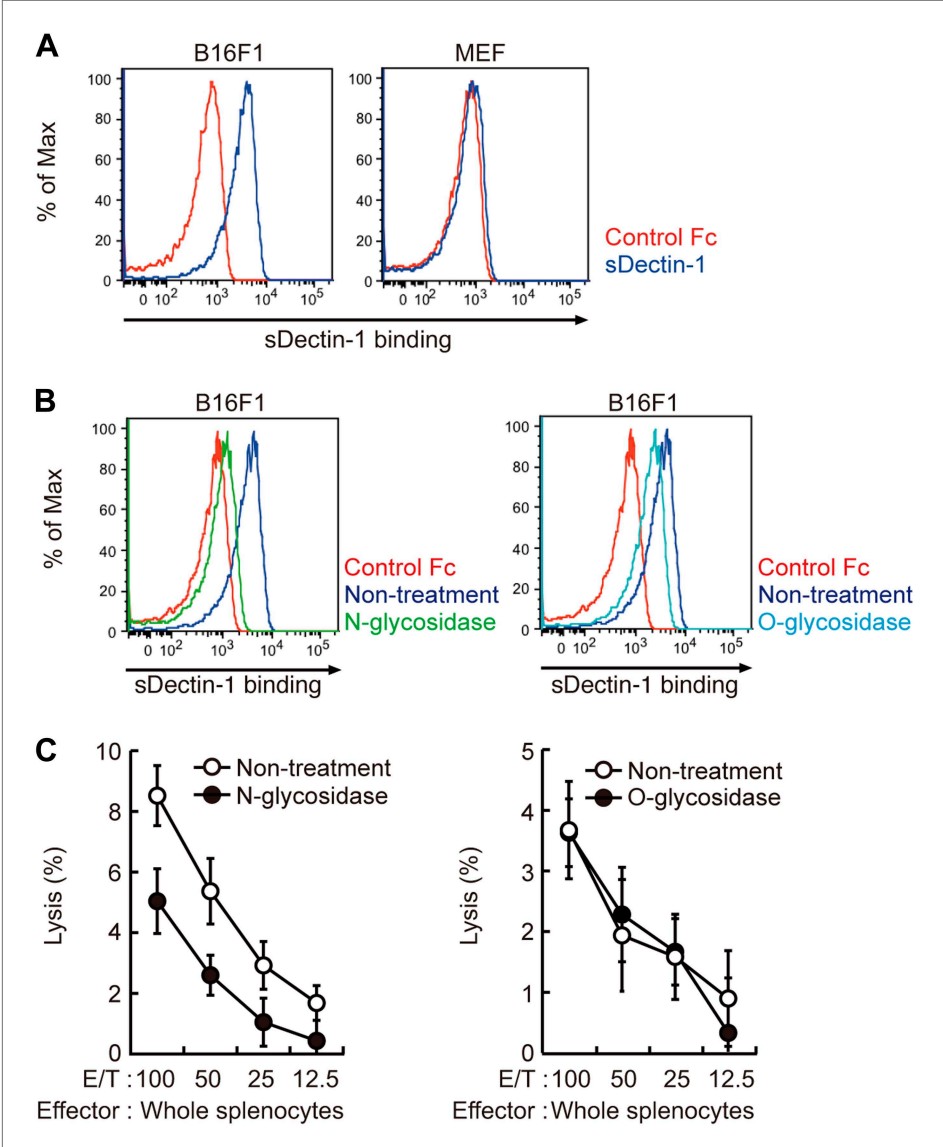

**Figure 3**. Recognition of N-glycan structures on B16F1 cells by Dectin-1 and its requirement for the enhancement of NK cell-mediated killing activity. (**A**) Binding of sDectin-1 to B16F1 cells (left panel) and primary mouse embryonic fibroblasts (MEFs; right panel). The cells (4 × 10$^5$ cells) were incubated with human IgG1 Fc (control Fc) or sDectin-1 (fused with the Fc) for flow cytometric analysis. (**B**) Effect of N-glycosidase treatment (left panel) or O-glycosidase in combination with neuraminidase (right panel) on the sDectin-1 binding to B16F1 cells. The cells (4 × 10$^5$ cells) were treated with either N-glycosidase (25 U/ml) or O-glycosidase (25 mU/ml) with neuraminidase (250 mU/ml) for 1 hr and then subjected to the sDectin-1 binding assay. These enzymatic reactions were performed under the conditions wherein these cells remain alive. (**C**) The effect of N-glycosidase or O-glycosidase treatment of B16F1 cells on in vitro killing activity of WT splenocytes. B16F1 cells were treated with or without N-glycosidase (25 U/ml) for 1 hr in RPMI medium (left panel) or treated with or without the combination of O-glycosidase (25 mU/ml) and neuraminidase (125 mU/ml) for 1 hr in RPMI medium (right panel) and then subjected to in vitro killing assay. $^{51}$Cr radioactivity released from target cells was measured. Represented as means ± SD. E/T: effector/target cell ratio. In in vitro killing assays, 1 × 10$^4$ of $^{51}$Cr-labeled B16F1 cells were used.

The following figure supplements are available for figure 3:

**Figure supplement 1**. Binding of sDectin-1 to various mouse primary cells.

**Figure supplement 2**. Effect of neuraminidase treatment on the sDectin-1 binding to B16F1 cells.

*Figure 3. Continued on next page*

*Figure 3. Continued*

**Figure supplement 3**. Effect of N-glycosidase treatment of B16F1 cells on in vitro killing activity of purified NK cells.

**Figure supplement 4**. Mass spectrometric analysis of N-glycosidase-treated B16F1 cells.

**Figure supplement 5**. Proposed N-glycan structures detected by mass spectrometric analysis.

**Figure supplement 6**. No differences in the amount of each N-glycan structure between samples with and without sDectin-1 treatment.

N-glycan structures, such as N-glycans with β1,6-GlcNAc branching (*Fernandes et al., 1991*), the expression of which is increased in tumor cells, were detected. Of note, pretreatment of the supernatant by sDectin-1 did not alter the MS peak pattern (*Figure 3—figure supplements 4–6*). This observation suggests that N-glycan structures, highly expressed in tumor cells, need to be bound to proteins for the Dectin-1 recognition (See below).

## Dectin-1 signaling and gene expression by tumor cells

To address the issue of whether Dectin-1 signaling is required for direct cell-to-cell contact or for the induction of a soluble mediator(s), we examined supernatants from CD11b$^+$ or CD11c$^+$ cells mixed with B16 cells and found that they had little, if any, effect on NK cell activity (*Figure 4A*). In addition, the use of antibodies against type I IFNs or IL12, both cytokines are known to activate NK cells (*Degli-Esposti and Smyth, 2005*), showed no effect on the NK cell-enhancing activity by DCs (*Figure 4B*). These results support the notion that Dectin-1 signaling is crucial for mediating cell-to-cell contact and subsequent NK activation.

We then examined the gene expression profiles of WT or Dectin-1-deficient DCs exposed to B16 cells by microarray analysis and found the induction of a substantial number of genes (*Figure 4C*; *Table 1*). The affected genes include several membrane-bound proteins amongst which *Inam* (termed as *Fam26f*) is known to activate NK cells via its homophilic interaction (*Ebihara et al., 2010*). Interestingly, the *Inam* mRNA induction by B16 cells is IRF5-dependent in splenic CD11c$^+$ cells (*Figure 4D*) and lentiviral expression of INAM cDNA in bone marrow-derived DCs resulted in the enhancement of NK cell-mediated killing of B16 cells (*Figure 4E*). Thus, the Dectin-1-IRF5-INAM pathway may participate in the DC-mediated activation of NK cells, at least in part, in this experimental setting. Obviously, more detailed analyses will be required to gain further mechanistic insight into the enhancement of NK-mediated tumor control by DCs and macrophages.

## Contribution of Dectin-1 to immune responses against other tumor cells

To what extent the Dectin-1-induced anti-tumor immune responses account for other types of tumor cells? To address this issue, we first examined the binding capacity of sDectin-1 for other tumor cell lines. The binding capacity was variable in that the binding was strong for 3LL (lung carcinoma), YAC-1 (lymphoma), and Meth-A (fibrosarcoma) cell lines, but was significantly weaker for some tumor cells such as SL4 (colon carcinoma) and B16F10 (melanoma) cell lines (*Figure 5A*, *Figure 5—figure supplement 1*). It is worth noting that the latter two cell lines are known to metastasize massively even in WT mice (*Morimoto-Tomita et al., 2005*) (See below).

In addition, chemical cross-linking of sDectin-1, followed by immunoblot analysis, generated multiple smear bands of similar pattern using B16F1 and 3LL cells, whereas bands were barely detectable with B16F10 cells or MEFs. Further, these bands are abolished by the pretreatment of cells with N-glycosidase (*Figure 5—figure supplement 2*). Thus, these data suggest that similar or identical N-glycan structures are expressed on multiple proteins that function as Dectin-1 ligands in these, and probably other, Dectin-1 binding tumor cells.

To examine the co-relationship between sDectin-1 binding and NK-mediated killing activity of splenocytes, we selected the lung carcinoma 3LL and YAC-1 cells with strong sDectin-1 binding and colon carcinoma SL4 cells with weak binding, and subjected them to the in vitro killing assay described above. We observed Dectin1-dependent cell killing activity against 3LL and YAC-1 cells (*Figure 5B*; left panel and *Figure 5—figure supplement 3*), whereas the killing activity of WT splenocytes against SL4 cells was significantly weaker as compared to that for 3LL or YAC-1 cells and was reduced only

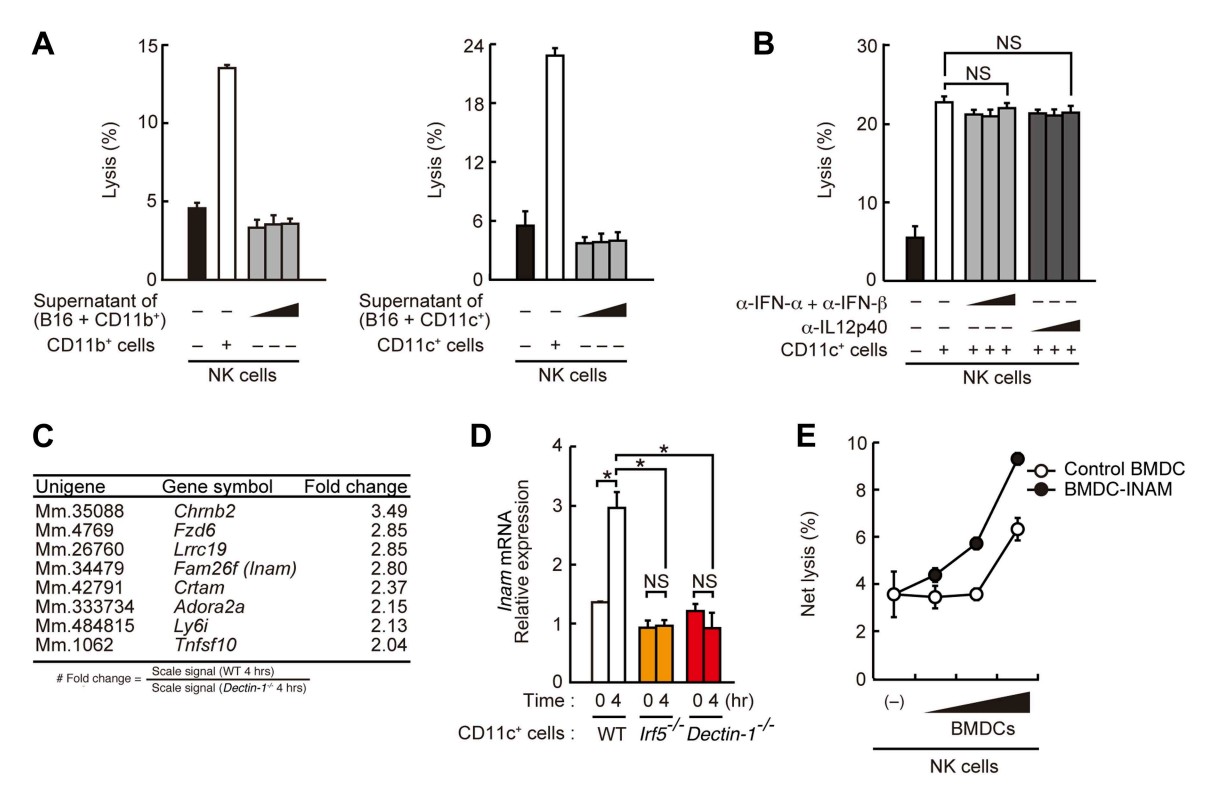

**Figure 4**. Requirement of cell-to-cell contact between NK cells and DCs for the enhancement of NK cell-mediated killing activity. (**A**) Effect of the supernatant of myeloid cells after incubation with tumor cells. NK cell killing activity against B16F1 cells was assessed in the presence of supernatants from the co-culture of B16F1 cells with splenic CD11b⁺ cells (left panel) or CD11c⁺ cells (right panel). Represented as means ± SD. (**B**) Effect of antibodies against cytokines on NK cell-mediated killing activity. In vitro killing activity of purified NK cells (WT; 1 × 10⁵ cells) against B16F1 cells in the presence of 3 × 10⁵ of splenic CD11c⁺ cells was assessed without or with neutralizing antibodies for type I IFNs or IL12p40. (**C**) Induction of mRNAs in DCs co-cultured with B16F1 cells via Dectin-1 signaling. WT or *Dectin-1⁻/⁻* splenic CD11c⁺ cells (3 × 10⁵ cells) were co-cultured with B16F1 cells (1 × 10⁴ cells). Total RNA from those cells was then isolated at time zero and 4 hr after the co-culture and then subjected to microarray analysis. We first identified genes for which mRNA is induced more than two-fold in WT DCs co-cultured with B16F1 cells and then, of those, selected the genes whose mRNA levels are increased more than twofold in WT 4 hr sample compared to *Dectin-1⁻/⁻* 4 hr sample. Those selected genes are listed in the order of fold change (WT 4 hr/*Dectin-1⁻/⁻* 4 hr) (**Table 1**). Genes encoding a membrane-bound protein are listed. (**D**) Induction of *Fam26f* (*Inam*) mRNA by the Dectin-1-IRF5 pathway. The expression levels of *Inam* mRNA were monitored by qRT-PCR analysis of total RNA from splenic CD11c⁺ cells (WT, *Irf5⁻/⁻*, or *Dectin-1⁻/⁻*; 3 × 10⁵ cells) co-cultured with B16F1 cells (1 × 10⁴ cells) for 4 hr as described in 'Materials and methods'. Results are presented relative to the expression of *Gapdh* mRNA. Represented as means ± SD. *p < 0.05 by Student's *t* test. NS, not significant. Although the IL15 cytokine system is known to promote growth and activity of NK cells, *Il15* and *Il15ra* mRNA expression levels were affected only marginally in the *Dectin-1⁻/⁻* DCs (**Figure 4—figure supplement 1**). The results suggest that this cytokine system will not be involved in this particular experimental setting. (**E**) Effect of INAM expression in DCs on the enhancement of NK cell killing activity. Purified NK cells (WT; 1 × 10⁵ cells) were mixed with increasing amounts (1 × 10⁴, 3 × 10⁴, and 1 × 10⁵ cells) of INAM-transduced WT BMDCs (BMDC-INAM) or mock-transduced WT BMDCs (control BMDC) and killing activities against B16F1 cells were monitored. Represented as means ± SD. In all in vitro killing assays, 1 × 10⁴ of ⁵¹Cr-labeled B16F1 cells were used. ⁵¹Cr radioactivity released from target cells was measured. The percentage of cytotoxicity was calculated as described in the legend of **Figure 1D** and represented as Net lysis (%).

The following figure supplement is available for figure 4:

**Figure supplement 1**. Induction of *Il15* or *Il15ra* mRNAs in DCs co-cultured with B16F1 cells.

marginally in Dectin-1-deficient splenocytes (**Figure 5B**; right panel). These results support the notion that tumor cells expressing Dectin-1 ligands at high levels are more susceptible to NK cell-mediated killing in a manner dependent on Dectin-1 signaling.

We further expended our in vivo analysis to 3LL and SL4 cells. As shown in **Figure 5C**, a marked enhancement of lung metastasis of 3LL cells was observed in Dectin-1-deficient mice, but not in WT mice (**Figure 5C**; left panel), while SL4 cells without significant sDectin-1 binding (**Figure 5A**; right

**Table 1.** Induction of mRNAs in DCs co-cultured with B16F1 cells via Dectin-1 signaling

| Unigene | Gene symbol | Fold change |
|---|---|---|
| Mm.275426 | Amy2a1///Amy2a2///Amy2a3///Amy2a4///Amy2a5 | 1995.76 |
| Mm.45316 | Cela2a | 173.36 |
| Mm.383263 | Try4///Try5 | 88.72 |
| Mm.475541 | Cela3b///Gm13011 | 61.79 |
| Mm.20407 | Pnlip | 21.28 |
| Mm.142731 | Reg1 | 16.65 |
| Mm.34374 | Ctrb1 | 16.42 |
| Mm.21160 | Clps | 14.85 |
| Mm.276926 | Prss2 | 13.67 |
| Mm.34692 | Cpb1 | 10.25 |
| Mm.450553 | Tpte | 9.99 |
| Mm.10753 | Pnliprp1 | 8.31 |
| Mm.46360 | Reg2 | 6.86 |
| Mm.1825 | Tff2 | 5.19 |
| Mm.867 | Ccl12///LOC100862578 | 4.96 |
| Mm.464256 | Tcl1b3 | 3.97 |
| Mm.2745 | Ctrl | 3.72 |
| Mm.35088 | Chrnb2 | 3.49 |
| Mm.14874 | Gzmb | 3.46 |
| Mm.159219 | Batf2 | 3.45 |
| Mm.439927 | Nol7 | 3.00 |
| Mm.4769 | Fzd6 | 2.85 |
| Mm.26760 | Lrrc19 | 2.85 |
| Mm.34479 | Fam26f (Inam) | 2.80 |
| Mm.4662 | Irg1 | 2.72 |
| Mm.4922 | Csf2 | 2.70 |
| Mm.2319 | Stmn3 | 2.70 |
| Mm.24375 | Ttpa | 2.70 |
| Mm.41416 | Rilp | 2.65 |
| Mm.159575 | Cdyl2 | 2.57 |
| Mm.261140 | Iigp1 | 2.44 |
| Mm.42791 | Crtam | 2.37 |
| Mm.26730 | Hp | 2.32 |
| Mm.203866 | Ahnak | 2.30 |
| Mm.34520 | Gtpbp8 | 2.30 |
| Mm.32368 | Krit1 | 2.29 |
| Mm.116997 | Hmmr | 2.26 |
| Mm.208125 | Adamts6 | 2.17 |
| Mm.333734 | Adora2a | 2.15 |
| Mm.484815 | Ly6i | 2.13 |
| Mm.271830 | Dhx58 | 2.10 |
| Mm.131098 | Golga1 | 2.10 |

*Table 1. Continued on next page*

panel) underwent massive metastasis in both WT and mutant mice (*Figure 5C*; right panel). Expectedly, massive metastasis of B16F10 cells was observed without significant difference between WT and Dectin-1-deficient mice (*Figure 5—figure supplement 4*), suggesting that B16F10 cells could have been selected to evade the Dectin-1-mediated anti-tumor orchestration. Thus, these in vivo observations are consistent with in vitro observations in that the recognition of and signaling by Dectin-1 in DCs and macrophages constitute a critical aspect of the NK cell-mediated anti-tumor innate immunity.

Finally, Dectin-1 binding to human tumor cell lines was examined by preparing a human sDectin-1. Interestingly, we found human sDectin-1 binds to several of these cell lines (*Figure 5—figure supplement 5*). We also observed the Dectin-1-dependent induction of *Inam* mRNA in mouse DCs by human cancer cell line HBC4, to which human sDectin-1 strongly binds (*Figure 5—figure supplement 6*). Since mouse and human Dectin-1 are highly conserved, we infer the mouse DCs respond to human Dectin-1 ligands for gene induction. As such, these results suggest the presence of a similar or identical anti-tumor Dectin-1 signaling mechanism in the human immune system.

## Discussion

The critical role of signal-transducing innate immune receptors in mediating innate and adaptive immune responses against invading pathogens is well known. However, it has been enigmatic if and how these receptors contribute to anti-tumor responses. In the present study, we demonstrated that the innate immune receptor Dectin-1 expressed on DCs and macrophages is important to NK-mediated killing of tumor cells. Our results indicate that NK cells are required to orchestrate with DCs and macrophages for cell killing, wherein activation of the IRF5 transcription factor by Dectin-1 signaling instigated by receptor recognition of N-glycan structures on tumor cells is critical. This notion is supported by an excessive growth of tumors with sDectin-1 binding in mice genetically deficient in either Dectin-1 or IRF5 in vivo. To our knowledge, this is the first demonstration that an innate immune receptor contributes to anti-tumor recognition and signaling through orchestration of innate immune cells. Our study offers new insight into the NK-mediated anti-tumor activity of the innate immune system (model depicted in *Figure 6*).

A more detailed structural analysis of the Dectin-1 ligand(s) needs to be clarified. Our

*Table 1. Continued*

| Unigene | Gene symbol | Fold change |
| --- | --- | --- |
| Mm.131723 | *Cxcl11* | 2.09 |
| Mm.5022 | *Mmp13* | 2.09 |
| Mm.291595 | *Klf9* | 2.09 |
| Mm.33691 | *Reg3d* | 2.08 |
| Mm.130 | *Socs1* | 2.06 |
| Mm.10948 | *Slfn1* | 2.05 |
| Mm.1062 | *Tnfsf10* | 2.04 |
| Mm.22213 | *Glipr2* | 2.01 |

The genes identified in the microarray analysis in *Figure 4C* are listed in the order of fold change (WT 4 hr/*Dectin-1*$^{-/-}$ 4 hr). Genes encoding a membrane-bound protein are indicated in red letters.

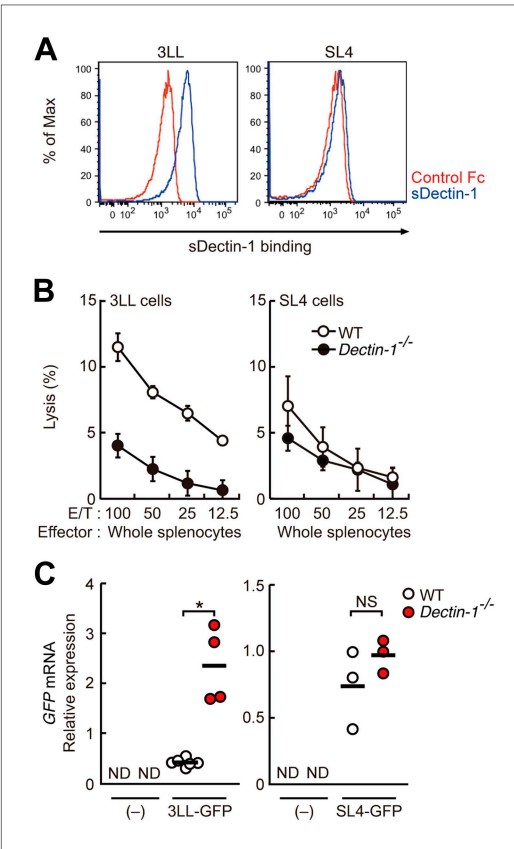

**Figure 5**. Dectin-1 binding on other tumor cells and its role in tumor suppression. (**A**) Binding of sDectin-1 to 3LL (left panel) or SL4 cells (right panel). The sDectin-1 binding to these cell lines was examined as described in *Figure 3A*. (**B**) In vitro killing activity of WT or *Dectin-1*$^{-/-}$ splenocytes against 3LL (left panel) or SL4 cells (right panel) as indicated E/T ratio. Represented as means ± SD. In in vitro killing assays, 1 × 10$^4$ of $^{51}$Cr-labeled target cells were used. (**C**) Quantification of metastasis of 3LL (left panel) or SL4 cells (right panel) in WT or *Dectin-1*$^{-/-}$ mice. WT and *Figure 5. Continued on next page*

sDectin-1 chemical cross-linking data suggest that similar or identical N-glycan structures are expressed on multiple protein molecules in tumor cells as Dectin-1 ligands. On the basis of the MS and chemical cross-linking data (*Figure 3—figure supplement 4–6*, *Figure 5—figure supplement 2*), we infer that N-glycan structures need to be bound to proteins for the Dectin-1 recognition.

Obviously further work will be required to discern the detailed nature of the ligands, it is possible that Dectin-1 recognition of and activation by N-glycan structures are contingent on the absolute expression levels of N-glycans and/or their associated proteins on the cell; that is, tumor cells with 'increased self' molecules are targets for innate recognition for activation of the immune system. However, it is also possible that Dectin-1 recognition requires particular, tumor-specific N-glycan-containing structures, which may belong to 'altered self' (*Medzhitov and Janeway, 2002*). Whichever the case, Dectin-1-binding structures may fit into the category of 'tumor-associated molecular patterns (TAMPs)' vis-à-vis PAMPs for invading pathogens and DAMPs for normal cells subjected to stress or death (*Rubartelli and Lotze, 2007*; *Kawai and Akira, 2010*). This issue obviously merits more advanced study.

Our data showing that Dectin-1-deficiency does not entirely abrogate the NK-enhancing activity of CD11b$^+$ or CD11c$^+$ cells suggests the interesting possibility that other CLRs are also involved in the orchestration of innate immune cells for the efficient NK cell-mediated anti-tumor immune response. It is curious that in many tumor cells glycosylation levels are generally enhanced, providing an advantage to tumor cells in migration and metastasis (*Granovsky et al., 2000*; *Gu and Taniguchi, 2008*). If so, the host's innate system could have evolved to limit the immune evasion of tumor development through recognition of glycosylation products via Dectin-1 and other CLR family members. As such, the potential involvement of other CLR members in the recognition tumor cells, to which sDectin-1 binding is low, is also an interesting future issue to be addressed.

The examination of the role of Dectin-1 and other CLRs in the control of tumors in de novo carcinogenesis models will be an exciting issue to address in future studies. It is interesting to note that Dectin-1 mRNA is also expressed in CD11b$^+$ or CD11c$^+$ cells from liver and lung (*Figure 2—figure supplement 3C*); hence, we infer that Dectin-1 signaling may contribute to the control of tumor progression in various tissues. In addition, since tumor cells are phenotypically and functionally heterogeneous even within the same

*Figure 5. Continued*

*Dectin-1⁻/⁻* mice were intravenously injected with GFP-expressing 3LL (3LL-GFP) or SL4 cells (SL4-GFP). *GFP* mRNA expression levels in lungs were assessed by qRT-PCR 12 (3LL cells) or 14 (SL4 cells) days after the injection. Means are indicated as black bars. *$p < 0.05$ by Student's *t* test; ND, none detected; NS, not significant.

The following figure supplements are available for figure 5:

**Figure supplement 1**. Binding of sDectin-1 to various mouse cancer cell lines.

**Figure supplement 2**. Pull-down analysis of sDectin-1 binding on cancer cells.

**Figure supplement 3**. In vitro killing activity of WT or *Dectin-1⁻/⁻* splenocytes against YAC-1 cells.

**Figure supplement 4**. Contribution of Dectin-1 signaling to anti-tumor killing activity against B16F10 cells.

**Figure supplement 5**. Binding of sDectin-1 to various human cancer cell lines.

**Figure supplement 6**. Induction of *Inam* mRNAs in DCs co-cultured with human cancer cells.

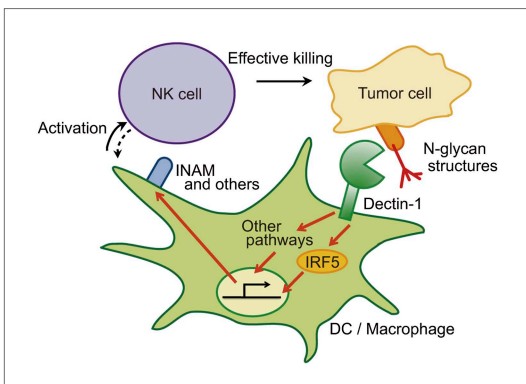

**Figure 6**. Schematic view of the orchestration of innate immune cells for NK cell-mediated tumor killing. Dectin-1 expressed by DCs and macrophages recognizes N-glycan structures on tumor cells and signals to activate IRF5 pathway and other pathways, thereby activating NK cells. Thus, NK cells require tumor recognition and signaling by these innate immune cells for their effective tumoricidal action. Although INAM is depicted here, the DC-mediated activation of NK cells would involve other molecules and this Dectin-1-IRF5-INAM pathway may represent only a part of the entire picture. It remains to be further characterized as to whether and how this and/or other effector ligands induced by Dectin-1 signaling contribute to NK activation in full. It is possible that DCs/macrophages are secondarily activated by yet unknown factors from NK cells (see dashed arrow).

tumor mass (*Meacham and Morrison, 2013*), it is possible that tumor cells show a differential expression profiles for Dectin-1 ligands in vivo. If so, it may be the case that the in vivo progression of a tumor is controlled via direct and indirect Dectin-1 signaling, that is, NK cells activated by CD11b⁺ or CD11c⁺ cells via Dectin-1 signaling may exert tumoricidal activities on tumor cells regardless on their Dectin-1 ligand expression. Obviously, these possibilities are speculative and need to be investigated further.

Identification of critical target molecules of Dectin-1 signaling that participate to the enhancement of NK cells for anti-tumor response is also an interesting, though unanswered question. Although our study suggests the involvement of INAM, we infer that other membrane molecules identified here (*Figure 4C*), which are without a known function at present, may also participate. This is obviously an important future issue to be rigorously addressed.

Our present study may also offer a novel view on the clinical efficacy of monoclonal antibodies (Abs) such as cancer therapeutics, specifically for how Ab-dependent cell-mediated cytotoxicity (ADCC) mediates antitumor cytotoxicity via Fc receptor(s) on immune cells (*Alderson and Sondel, 2011*; *Seidel et al., 2013*). Although the conventional wisdom has been that Fc receptors (FcRs) expressed on NK cells are responsible for ADCC, primarily based on in vitro studies using human mononuclear cells (*Alderson and Sondel, 2011*; *Seidel et al., 2013*), it has been appreciated for some time that in the mouse, Ab mediated tumor killing is mediated primarily by myeloid cells. For example, mouse IgG2a is the most potent Ab isotype for the activation of effector cells through its intermediate–affinity interaction with FcRIV (*Nimmerjahn and Ravetch, 2005*; *Nimmerjahn et al., 2010*). Yet, this FcR is not expressed by NK cells, but is expressed by DCs and macrophages (*Nimmerjahn et al., 2005*). It is also interesting to note that FcRIV utilizes FcRγ, the FcR common γ subunit, to transmit the Ab signal to the cell interior via pathways similar, if not identical, to those of CLRs (*Honda and Taniguchi, 2006*). As such, Abs may activate DCs and macrophages via FcRIV-FcRγ pathway to mediate both ADCC and further contribute to NK cell activation through a mechanism similar to the pathway we describe in this study. The process may be termed 'indirect ADCC'. Clearly, Ab-mediated cancer therapeutics likely involves multiple mechanisms (*Simpson et al., 2013*); therefore, our speculation obviously needs further investigation, particularly in the context of human cancer therapeutics.

Finally, in light of the tenet that innate immunity instructs adaptive immunity (*Janeway and Medzhitov, 2002*), our study may also raise an interesting issue of whether Dectin-1 signaling by tumor recognition also affects adaptive anti-tumor immune responses, and also may provide new means for the efficient immune responses for cancers such as Dectin-1 agonistic Abs.

## Materials and methods

### Mice

C57BL/6 mice were from CLEA Japan Inc. *Clec7a (Dectin-1)$^{-/-}$*, *Irf3$^{-/-}$*, *Irf5$^{-/-}$*, *Irf7$^{-/-}$*, *Rag1$^{-/-}$*, and *Myd88$^{-/-}$* mice as described previously were maintained on a C57BL/6 genetic background (*Sato et al., 2000*; *Honda et al., 2005*; *Takaoka et al., 2005*; *Honda and Taniguchi, 2006*; *Saijo et al., 2007*; *Kano et al., 2008*). CD11c-DTR and C57BL/6-Ly5.1 mice were obtained from The Jackson Laboratory (Bar Harbor, ME). Each experiment was performed using sex-, age-matched mice. All animal care and experiments conformed to the guidelines for animal experiments of the University of Tokyo, and were approved by the animal research committee of the University of Tokyo (Reference number: P10-122 and P10-123).

### Reagents

Curdlan was purchased from Wako Chemicals (Japan). FK506 was from Enzo Life Sciences (Farmingdale, NY). N-glycosidase, O-glycosidase, and neuraminidase were from Roche Diagnostics (Switzerland). Anti-IFN-α (4E-A1) and anti-IFN-β (7F-D3) monoclonal antibodies were purchased from YAMASA (Japan). Anti-IL12p40 antibody (C17.8), APC-conjugated anti-CD49b antibody (DX5), APC-conjugated anti-CD11b antibody (M1/70), APC-conjugated anti-CD11c antibody (N418), and FITC-conjugated anti-F4/80 antibody (BM8) were from Biolegend (San Diego, CA). FITC-conjugated anti-CD3ε antibody (145-2C11) was from BD Biosciences (San Jose, CA).

### Cells

Mouse melanoma cell line B16F1 cells and B16F10 cells, Lewis lung carcinoma cell line 3LL cells, fibro-sarcoma cell line Meth-A cells, and HEK293T cells were obtained from RIKEN BioResource Center (Japan). Mouse colon carcinoma cell line SL4 cells were kindly provided by Dr T Irimura (The University of Tokyo). B16F1 cells, B16F10 cells, and HEK293T cells were maintained in Dulbecco's Modified Eagle's Medium (DMEM; Nissui, Japan) supplemented with 10% fetal calf serum (FCS; Hyclone, Logan, UT). SL4 cells were cultured in DMEM-F12 medium (Gibco/Invitrogen, Grand Island, NY) supplemented with 10% FCS. 3LL cells and Meth-A cells were cultured in RPMI medium (Nacalai Tesque, Japan) supplemented with 10% FCS. Mouse lymphoma cell line YAC-1 cells were cultured as described previously (*Sato et al., 2001*). Primary mouse embryonic fibroblasts were isolated and cultured as described previously (*Yanai et al., 2009*).

Human hepatocellular carcinoma cell line HepG2 cells and melanoma cell line G-361 cells were obtained from National Institute of Biomedical Innovation (Japan). Human cervical carcinoma cell line HeLa cells, lung adenocarcinoma cell line A549 cells, colorectal carcinoma cell line HCT116 cells, gastric carcinoma cell line MKN45 cells, prostate cancer cell line PC-3 cells, and chronic myelogenous leukemia cell line K562 cells were obtained from RIKEN BioResource Center. Human breast cancer cell line HBC4 cells were kindly provided by Dr K Yamazaki (Cancer Institute Hospital of JFCR, Japan). Human hepatoma cell line Huh7 cells were kindly provided by Dr M Kohno (Toray Co.). Human glioblastoma cell lines T98 cells and U251 cells were kindly provided by Dr J Yoshida (Nagoya University). HepG2 cells, Huh7 cells, HeLa cells, T98 cells, A549 cells, and HCT116 cells were maintained in DMEM supplemented with 10% FCS. HBC4 cells, MKN45 cells, PC-3 cells, and K562 cells were cultured in RPMI medium supplemented with 10% FCS. U251 cells were maintained in DMEM supplemented with 10% FCS and non-essential amino acids (Nacalai Tesque). G-361 cells were cultured in McCoy's 5A medium (Gibco/Invitrogen) supplemented with 10% FCS. Mycoplasma contaminations of cell lines were checked by MycoAlert Mycoplasma Detection Kit (Lonza, Switzerland) according to manufacturer's instruction; no evidence of contamination was found.

To prepare splenocytes, spleens were digested with collagenase D (Roche Diagnostics) and DNase I (Roche Diagnostics) as described previously (*Negishi et al., 2013*). Splenic CD11b$^+$, CD11c$^+$, B, or T cells were purified using CD11b or CD11c MicroBeads, B cell isolation kit, or pan T cell isolation kit II, respectively (Miltenyi Biotec, Germany). For CD8$^+$CD11c$^+$ or CD8$^-$CD11c$^+$ cell purification, cells were negatively selected from splenocytes using anti-mouse CD90.2 (30-H12; eBioscience, San Diego, CA),

anti-mouse CD5 (53-7.3; eBioscience), and anti-mouse B220 (RA3-6B2; eBioscience) antibodies with Dynabeads M-280 sheep anti-mouse IgG (Invitrogen), and CD19 and CD49b (DX5) MicroBeads (Miltenyi Biotec). CD8$^+$CD11c$^+$ and CD8$^-$CD11c$^+$ cells were selected using CD8a (Ly-2) and CD11c MicroBeads (Miltenyi Biotec). CD11c$^-$CD11b$^+$ cells were purified using CD11b Microbeads from splenocytes of CD11c-DTR mice after depletion of CD11c$^+$ cells by intraperitoneal injection of diphtheria toxin (*Jung et al., 2002*) (4 ng/g body weight; Sigma, St Louis, MO). For NK cells purification, splenocytes were incubated within nylon-wool column (Wako Chemicals) for 1 hr, and then NK cells were negatively isolated from the nylon-non-adherent cells with NK cell isolation kit II (Miltenyi Biotec) as described previously (*Takeda et al., 2011*). For NK cell depletion, mice were injected intraperitoneally with 200 µg anti-asialo GM1 antibody (Wako Chemicals). For liver cell and lung cell preparation, livers and lungs were digested with collagenase D and DNase I, passed through a sterile 40 µm pore size nylon cell strainer (BD Biosciences) and the cell suspensions were treated with RBC lysis buffer (eBioscience). To purify CD11b$^+$ and CD11c$^+$ cells from livers and lungs, the cell were stained with APC-conjugated anti-CD11b antibody and FITC-conjugated anti-F4/80 antibody or APC-conjugated anti-CD11c antibody. CD11b$^+$F4/80$^+$ cells and CD11c$^+$ cells were sorted using FACSAria (BD Biosciences). To generate bone marrow-derived dendritic cells (BMDCs), bone marrow cells were cultured with 20 ng/ml mouse GM-CSF (BD Pharmingen, San Diego, CA) in RPMI medium supplemented with 10% FCS for 6 days.

## Lung metastasis assay

Mice were intravenously injected with $1 \times 10^6$ or $2 \times 10^6$ of B16F1 cells, $1 \times 10^6$ of 3LL-GFP cells, $3 \times 10^5$ of SL4-GFP cells, or $5 \times 10^5$ of B16F10 cells. For B16F1 cells and B16F10 cells, the number of metastatic colonies in lungs was counted after 14 days of injection. Lungs were harvested 12 (3LL-GFP cells) or 14 (SL4-GFP cells) days after injection. Total RNA was purified from lungs and subjected to quantitative reverse transcription PCR (qRT-PCR) to measure *GFP* mRNA expression level. GFP-expressing 3LL cells and SL4 cells (3LL-GFP cells and SL4-GFP cells) were established using retrovirus-mediated gene transfer system as described previously (*Nakagawa et al., 2005*). pMX-GFP retrovirus expression vector was kindly provided by Dr T Kitamura (The University of Tokyo).

## Subcutaneous tumor growth assay

Mice were injected subcutaneously with $1 \times 10^5$ of B16F1 cells. Tumor volume was evaluated using the equation volume = $\pi ab^2/6$, where a and b are the lengths of the major and minor axes, respectively (*Ikushima et al., 2008*).

## Bone marrow transplantation

Bone marrow transplantation was performed as described previously (*Honda et al., 2004*). Briefly, recipient mice were subjected to sublethal γ-irradiation (5 Gy × 2 with a 3-hr interval), and were injected with $5 \times 10^6$ of bone marrow cells freshly isolated from donor mice. 3 months after the transplantation mice were subjected to metastasis assay. All mice showed a high-degree chimerism (90–95%).

## In vitro killing assay

Killing activity of NK cells was tested by $^{51}$Cr release assay (*Takeda et al., 2011*). Splenocytes or NK cells were prepared from WT, *Irf5$^{-/-}$*, or *Dectin-1$^{-/-}$* mice and used as effector cells. Target cells suspended in RPMI medium were incubated with 100 µCi Na$_2$$^{51}$CrO$_4$ (Perkin Elmer, Waltham, MA) for 1 hr at 37°C and washed with PBS (Nacalai Tesque) three times. Target cells were then mixed with effector cells at the indicated effector/target (E/T) ratios in flat-bottomed 96-well plates with a total volume of 200 µl and then co-cultured for 4 hr at 37°C. In all in vitro killing assays, $1 \times 10^4$ of $^{51}$Cr-labeled target cells were used. Released $^{51}$Cr radioactivity in the supernatants was measured with a Wallac 1480 Wizard 3″ gamma counter (Perkin Elmer). The percentage of cytotoxicity was calculated as following formula: specific lysis (%) = (experimental release − spontaneous release)/(maximum release − spontaneous release) × 100. Target cell lysis was also measured by co-culturing target cells with myeloid cells in the absence of NK cells. This background value was subtracted from the total values obtained in the presence of NK cells and the calculated percentage of cytotoxicity was represented as Net lysis (%). All experiments were performed at least three times and the results were highly reproducible. We also confirmed there was no discernible phagocytosis by myeloid cells in our killing assays.

## Immunoblotting assay

Immunoblotting assay was performed as described previously (*Yanai et al., 2011*). Antibodies for IRF5 and USF-2 were purchased from Cell Signaling (Danvers, MA) and Santa Cruz (Santa Cruz, CA), respectively. USF-2 was used as a nuclear marker protein.

## qRT-PCR

Total RNA from tissues or cells was extracted using RNAiso (TAKARA, Japan) or NucleoSpin RNA II (MACHEREY NAGEL, Germany), and was reverse-transcribed with PrimeScript RT Master Mix (TAKARA). qRT-PCR was performed on Light Cycler 480 (Roche Bioscience) using the SYBR Green PCR Master Mix (Roche Bioscience) and values were normalized to the expression of *Gapdh* mRNA as described previously (*Yanai et al., 2009*). Primer sequences are as follows: *Gapdh* forward 5′-ctcatga ccacagtccatgc-3′; *Gapdh* reverse 5′-cacattgggggtaggaaacac-3′; *Clec7a* forward 5′-catcgtctcaccgta ttaatgcat-3′; *Clec7a* reverse 5′-cccagaaccatggccctt-3′; *Gzmb* forward 5′-accaaacgtgcttcctttcg-3′; *Gzmb* reverse 5′-tttggtgaaagcacgtggag-3′; *Prf1* forward 5′-tctccccactctggtttcca-3′; *Prf1* reverse 5′-gagat ggggcagacacttgg-3′; *Ifng* forward 5′-tggctttgcagctcttcctc-3′; *Ifng* reverse 5′-tccttttgccagttcctcca-3′; *Il6* forward 5′-acgatgatgcacttgcagaa-3′; *Il6* reverse 5′-gtagctatggtactccagaagac-3′; *Tnf* forward 5′-tcatacc aggagaaagtcaacctc-3′; *Tnf* reverse 5′-gtatatgggctcataccagggttt-3′; *Fam26f* forward 5′-gacacagttggcc gaagaga-3′; *Fam26f* reverse 5′-aacgctgagatttcctgcca-3′; *GFP* forward 5′-cttcttcaagtccgccatgc-3′; *GFP* reverse 5′-gtgtcgccctcgaacttcac-3′; *Il15* forward 5′-catccatctcgtgctacttgtg-3′; *Il15* reverse 5′-gcctctg ttttagggagacct-3′; *Il15ra* forward 5′-gctgacatccgggtcaagaa-3′; *Il15ra* reverse 5′-cacttgaggctgggag ttgt-3′. All experiments were performed at least three times in triplicate.

## ELISA

Purified NK cells (WT or *Dectin-1*$^{-/-}$; $5 \times 10^4$ cells) were stimulated with 10 or 100 ng/ml of recombinant mouse IL12 (R&D Systems, Minneapolis, MN). IFN-γ production in culture supernatants was evaluated by ELISA kit (R&D Systems) after 24-hr incubation.

## MCMV preparation and infection

Mouse cytomegalovirus (MCMV; Smith strain) was purchased from ATCC. MCMV stocks were prepared as described (*Brune et al., 2001*). Briefly, MEFs were infected with MCMV and incubated for 72 hr until the cell layer was completely infected. Then, the cells and supernatant were collected and were subjected to three freeze-and-thaw cycles to release cell-associated virus. The crude virus-containing medium was centrifuged and filtrated with a sterile 0.45 μm pore size filter (Millipore, Billerica, MA). MCMV in the filtrated medium was concentrated by centrifugation at $26,000 \times g$ (15,000 rpm; Beckman SW 41 Ti rotor) for 3 hr at 4°C. The pellet was resuspended in PBS and filtrated through a sterile 0.45 μm filter. MCMV titer was determined by plaque-forming cell assay. MEFs were infected with serially diluted ($\log_{10}$ steps) virus stock solution or organ homogenates and incubated for 2 hr at 37°C. Then, the cells were washed extensively and cultured in 2.4% methylcellulose-containing DMEM supplemented with 4% FCS for 5 days. After incubation, cells were fixed with formalin and stained with crystal violet solution. Virus plaques were then counted. WT and *Dectin-1*$^{-/-}$ mice were intraperitoneally infected with MCMV ($5 \times 10^3$ or $3 \times 10^5$ pfu). For NK depletion, WT mice were given anti-asialo GM1 antibody intravenously one day before infection. Mice were sacrificed at 3 days postinfection and spleens from the mice were excised and homogenized in PBS.

## Preparation of a soluble form of Dectin-1 (sDectin-1) protein

The DNA fragment encoding extracellular domain (amino acid residues 73–244) of murine Dectin-1 was cloned using forward primer 5′-aaagatcttacccatacgatgttccagattacgctaattcagggagaaatc-3′ and reverse primer 5′-aaaaatctagacagttccttctcacagatac-3′, and inserted into the *Bgl*II sites of pFUSE-hIgG1-Fc2 vector (Invivogen, San Diego, CA), to make HA-tagged murine sDectin-1 (msDectin-1) conjugated with human IgG1 Fc region (*Hino et al., 2012*). The DNA fragment encoding extracellular domain (amino acid residues 73–247) of human Dectin-1 was cloned using forward primer 5′-agatcttacccatacga tgttccagattacgctaattcaggaagcaacacattgg-3′ and reverse primer 5′-agatctcattgaaaacttcttctcac-3′ to make HA-tagged human sDectin-1 (hsDectin-1) conjugated with human IgG1 Fc region. HEK293T cells were transfected with pFUSE-hIgG1-Fc2-HA-msDectin-1, pFUSE-hIgG1-Fc2-HA-hsDectin-1, or pFUSE-hIgG1-Fc2 empty vector using X-tremeGENE 9 DNA Transfection Reagent (Roche Applied Science). Then, supernatants were filtrated and gently mixed with Protein A Sepharose Fast Flow

(GE Healthcare, Waukesha, WI). The mouse sDectin-1, human sDectin-1, or human IgG1 Fc (control Fc) was eluted with 100 mM Glycine-HCl (pH 3.0), dialyzed with PBS, and concentrated using Amicon Ultra centrifugal filter (Millipore).

## Flow cytometric assay

Cells were incubated with sDectin-1 or human IgG1 Fc in TBS (pH 8.0) containing 1.3 mM $CaCl_2$ for 15 min at 4°C, and then labeled with anti-human IgG1 antibody (4E3; Abcam, Cambridge, MA) conjugated with allophycocyanin (APC). The cells were then analyzed by LSRII/Fortessa (BD Biosciences). APC conjugation to anti-human IgG1 antibody was performed using APC Labeling Kit-$NH_2$ (Dojindo, Japan) according to the manufacturer's protocol.

## Glycan analysis

After B16F1 cells ($1 \times 10^7$ cells) were treated with N-glycosidase (25 U/ml) for 1 hr, the supernatant was collected and incubated at 75°C for 15 min to inactivate N-glycosidase. Then, the supernatant was treated with or without sDectin-1 and Protein G Sepharose 4 Fast Flow (GE Healthcare). N-glycans remained in the supernatant were analyzed by Glycan*MAP* (Ezose Science, Pine Brook, NJ) (*Furukawa et al., 2008*).

## Microarray analysis

Microarray analysis was performed as described previously (*Negishi et al., 2012*). WT or *Dectin-1*$^{-/-}$ splenic CD11c$^+$ cells ($3 \times 10^5$ cells) were co-cultured with B16F1 cells ($1 \times 10^4$ cells) for 4 hr. Total RNA was extracted from co-cultured B16F1 cells and CD11c$^+$ cells and analyzed by GeneChip Mouse Genome 430 2.0 Array (Affymetrix, Santa Clara, CA).

## Lentivirus-mediated gene transfer

pCSII-EF-MCS-IRES2-Venus (pCSII-EF), pMDLg/pRRE, and pCMV-VSV-G-RSV-Rev vector were kindly provided by Dr H Miyoshi (RIKEN). Mouse INAM cDNA was cloned by PCR using primers (Forward 5′-gtcgac atggaaaagttcaaggcagtg-3′, Reverse 5′-gcggccgctcatagttcgtgagtgttagtcat-3′) and inserted into pT7-Blue T-vector (TAKARA). The cDNA was excised by *XhoI* and *NotI* digestion, blunted, and inserted into the blunted *NotI* site of pCSII-EF-MCS-IRES2-Venus vector (pCSII-EF-INAM). A total of $5 \times 10^6$ HEK293T cells were seeded on 10-cm dishes 24 hr prior to transfection. pCSII-EF (6 µg) or pCSII-EF-INAM (6 µg) vector in combination with pMDLg/pRRE (6 µg) and pCMV-VSV-G-RSV-Rev (6 µg) vectors was co-transfected into HEK293T cells. The medium was replaced after 4 hr. The lentivirus-containing medium was collected after another 48 hr, cleared by low-speed centrifugation, and filtrated through 0.45 µm filter. Virus particles were concentrated by ultracentrifugation at 70,000×$g$ (25,000 rpm; Beckman SW 41 Ti rotor) for 90 min at 4°C. The pellet was resuspended in RPMI supplemented with 10% FCS and filtrated through a sterile 0.45 µm filter, and the viral solution was subdivided to store at −80°C. The transduction efficiency was assessed by fluorescence of Venus ligated in the lentivirus vector, and the frequency of positive cells determined by flow cytometry in comparison with that of non-infected cells. Briefly, serially diluted viral solution was added to BMDCs ($3 \times 10^5$ cell/500 µl of RPMI medium supplemented with 10% FCS with 4 µg/ml polybrene [Sigma] in 24-well plate), centrifuged at 1200×$g$ (2360 rpm; TOMY LX-141) for 90 min at 30°C, and incubated for 3 hr. These cells were cultured for 20 hr additionally in fresh medium after wash. Then, the fluorescence intensities of the cells were analyzed by flow cytometry. The appropriate viral titer was determined so as to reach 90% Venus expression in BMDCs and was used for the experiments.

## Pull-down of Dectin-1 ligand

B16F1 and 3LL cells ($4 \times 10^7$ cells) treated with or without N-glycosidase (25 U/ml) and B16F10 cells and MEFs ($4 \times 10^7$ cells) treated without N-glycosidase were incubated with sDectin-1. After the unbound sDectin-1 was washed out with PBS, sDectin-1 and its ligand on these cells were cross-linked by Sulfo-NHS-LC-Diazirine (Thermo Scientific, Waltham, MA) according to manufacturer's instruction. Cells were lysed in T-PER Tissue Protein Extraction Reagent (Thermo Scientific), followed by the immunoprecipitation with Protein G Sepharose 4 Fast Flow (GE Healthcare). After elution with 100 mM Glycine-HCl (pH 3.0), sDectin-1-ligand complex was detected by Immunoblotting assay using HA-probe (F-7) HRP (Santa Cruz Biotechnology).

## Statistical analysis

Data are expressed as mean ± SD Student's *t* test was performed and difference was considered to be statistically significant when p-value < 0.05.

## Acknowledgements

We thank Drs N Taniguchi, K Hatanaka, Y Yamaguchi, S Kitazume, K Ogasawara, and A Takaoka for the helpful comments. We also thank Ms K Adachi, Mr M Taniguchi, Mr K Nishimura, Mr A Matsuda, Drs H Asada, and T Mizutani for technical assistance. This work was supported in part by a Grant-In-Aid for Scientific Research on Innovative Areas from the Ministry of Education, Culture, Sports, Science, and Technology of Japan, and by Core Research for Evolutional Science and Technology (CREST) of the Japan Science and Technology Agency (JST). SC, HU, YK, and SH are research fellows of the Japan Society for the Promotion of Science. The Department of Molecular Immunology is supported by BONAC Corporation and Kyowa Hakko Kirin Co., Ltd.

## Additional information

### Competing interests

TTaniguchi: Senior editor, *eLife*. The other authors declare that no competing interests exist.

### Funding

| Funder | Author |
| --- | --- |
| Japan Society for the Promotion of Science | Tadatsugu Taniguchi |
| Core Research for Evolutional Science and Technology, Japan Science and Technology Agency | Hideyuki Yanai, Junko Nishio, Hideo Negishi, Tadatsugu Taniguchi |

The funders had no role in study design, data collection and interpretation, or the decision to submit the work for publication.

### Author contributions

SC, HI, HU, HY, JN, Conception and design, Acquisition of data, Analysis and interpretation of data, Drafting or revising the article; YK, Acquisition of data, Analysis and interpretation of data, Drafting or revising the article; SH, Acquisition of data, Drafting or revising the article; HN, Analysis and interpretation of data, Drafting or revising the article; TTamura, SS, YI, Conception and design, Drafting or revising the article; TTaniguchi, Conception and design, Analysis and interpretation of data, Drafting or revising the article

### Ethics

Animal experimentation: All animal care and experiments conformed to the guidelines for animal experiments of the University of Tokyo, and were approved by the animal research committee of the University of Tokyo (Reference number: P10-122 and P10-123).

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
