## [Decision Letter]

Thank you for sending your work entitled “Recognition of tumor cells by Dectin-1 orchestrates innate immune cells for anti-tumor responses” for consideration at *eLife*. Your article has been favorably evaluated by K VijayRaghavan (Senior editor) and 3 reviewers, two of whom are members of our Board of Reviewing Editors.

The Reviewing editor was Ruslan Medzhitov. The other two reviewers have agreed to reveal their identity and were Jeffrey Ravetch and Xuetao Cao.

We are very happy to see this accepted for publication by *eLife*. However, in a resubmission please address, where appropriate, the minor comments of the reviewers:

1) The authors show that Dectin-1 in DC is required for NK cell-mediated anti-tumor activity, and Dectin-1 is dispensable for production of IFN-γ by purified NK cells alone. It's not shown whether Dectin-1 signal in DC is involved in NK cell expression of inflammatory and cytotoxic mediators in the co-culture system.

2) Did the authors detect the regulation of NK cell-mediated tumor cell lysis by DC in a transwell system?

3) Could the authors show the binding between Dectin-1 on DC and N-glycan on tumor cells in confocal images?

4) Liver is endowed with regulatory immune function. It is well known that liver is enriched in NK cells, and the chronic inflammation (especially virus infection-induced inflammation) may be involved in the pathogenesis of hepatocellular carcinoma. Do Kupffer cells express Dectin-1? If so, does Dectin-1 in Kupffer cells affect NK cell-mediated anti-tumor responses? The authors should discuss this possible application of the findings of innate receptors in the control of liver cancer.

5) Upon activation by DC recognition of N-glycan structures in tumor via Dectin-1, do the activated NK cells selectively kill the N-glycan-expressing cells of same origin? Could the authors discuss this?

---

## [Author Response]

*1) The authors show that Dectin-1 in DC is required for NK cell-mediated anti-tumor activity, and Dectin-1 is dispensable for production of IFN-γ by purified NK cells alone. It's not shown whether Dectin-1 signal in DC is involved in NK cell expression of inflammatory and cytotoxic mediators in the co-culture system*.

We agree with the reviewers that this is an interesting issue to address further. Accordingly, we examined mRNA expression levels for typical inflammatory and cytotoxic mediators in the co-culture system, wherein the ratio of B16F1 cells, DCs, and NK cells is 1:30:10. Total RNA was isolated at zero time and 4 hrs after the co-culture and subjected to qRT-PCR analysis. We found that the mRNA levels for inflammatory cytokines, namely, IFN-g, IL-6, TNF-a remained unchanged between the co-culture with WT DCs and Dectin-1-deficient DCs. Moreover, we did not find any differences in mRNA expression for cytotoxic mediators such as granzyme B and perforin-1. These results are now presented as Figure 2—figure supplement 8. As such, from these analyses, it seems unlikely that Dectin-1 signaling in DCs affects the expression of these molecules in NK cells. We agree that how NK cells are activated by DCs and macrophages that received Dectin-1 signaling to exert their anti-tumor cytotoxicity is a very interesting issue, but hope that the reviewer will agree that this out of the main scope of our present study.

2) Did the authors detect the regulation of NK cell-mediated tumor cell lysis by DC in a transwell system?

We have not examined the DC effect on NK cells in the transwell system. However, two experiments were carried out to examine the potential involvement of soluble mediators in the DC-mediated activation of NK cells. First, we examined supernatants from CD11b^+^ or CD11c^+^ cells mixed with B16 cells and found that they essentially had no effect on NK cell activity (Figure 4). Second, antibodies against type I IFNs or IL12, both known as the potent cytokines for NK cell activation, showed no effect on the NK cell-enhancing activity by CD11c^+^ cells (Figure 4). Taken together, these results support the notion that Dectin-1 signaling is crucial for mediating cell-to-cell contact and subsequent NK activation. In addition, we found no difference between WT and Dectin-1^-/-^ DCs in the mRNA induction levels for IL15 or IL15Ra (Figure 4—figure supplement 1). As such, we think it is most likely that Dectin-1 signaling is crucial for mediating cell-to-cell contact between DCs/macrophages and tumor cells and that this contact leads to NK activation. However, we cannot exclude the possibility that DCs/macrophages are secondarily activated by yet unknown factors from NK cells in the actual co-culture situation. Therefore, in order to make this point clearer, we revised the summary figure (Figure 6), describing the possibility of bidirectional activation between DCs/macrophages and NK cells. This point is also mentioned in the legend of Figure 6.

3) Could the authors show the binding between Dectin-1 on DC and N-glycan on tumor cells in confocal images?

We agree that this is potentially a very interesting experiment. However, we find that the following two elements make such an experiment difficult. First, N-glycan structures on tumor cells are likely to be diverse (see Figure 3—figure supplement 4, Figure 3—figure supplement 5 and Figure 3—figure supplement 6) and a suitable tool antibody is essentially not available. Second, Dectin-1 *per se* is also an N-glycosylated protein. As such, although interesting, we hope that the reviewers agree that we cannot perform this experiment.

*4) Liver is endowed with regulatory immune function. It is well known that liver is enriched in NK cells, and the chronic inflammation (especially virus infection-induced inflammation) may be involved in the pathogenesis of hepatocellular carcinoma. Do Kupffer cells express Dectin-1? If so, does Dectin-1 in Kupffer cells affect NK cell-mediated anti-tumor responses? The authors should discuss this possible application of the findings of innate receptors in the control of liver cancer*.

In response to these comments, we isolated macrophages and DCs from liver and lung, and examined the mRNA expression levels for Dectin-1 by qRT-PCR, as we previously examined in these cells from the spleen. Perhaps expectedly, *Dectin-1* mRNA expression was also observed in these cells, and these new data is now presented in Figure 2—figure supplement 3. Therefore, it is likely that Dectin-1 signaling in DCs and macrophages contributes to control of tumor progression in various tissues including lung and liver, as the reviewer pointed out. We discuss this point in the light of this data in the text as follows “we infer that Dectin-1 signaling may contribute to the control of tumor progression in various tissues.”

5) Upon activation by DC recognition of N-glycan structures in tumor via Dectin-1, do the activated NK cells selectively kill the N-glycan-expressing cells of same origin? Could the authors discuss this?

This is a very interesting point, although difficult to address experimentally. We think it is possible that once NK cells are activated by DCs/macrophages via Dectin-1 signaling, these NK cells will exert more potent tumoricidal activities on other tumor cells that do not express Dectin-1 ligands. Since it has been shown that tumor cells are phenotypically and functionally heterogeneous even in a given tumor mass (Meacham CE and Morrison SJ, *Nature*, 501, 328–337, 2013), it is possible that tumor cells show different expression profiles with each other for Dectin-1 ligands *in vivo*. If so, an interesting possibility is that those tumor cells, expressing Dectin-1 ligands, effectively induce DC/macrophage-mediated NK cell activation and that these so-called “licensed” NK cells may exert tumoricidal activities on tumor cells regardless on their Dectin-1 ligand expression. We thank the reviewers for pointing this out and we now discuss this possibility in the Discussion.